# Antibiotic Use in Dental Implant Procedures: A Systematic Review and Meta-Analysis

**DOI:** 10.3390/medicina59040713

**Published:** 2023-04-05

**Authors:** Elham Torof, Hana Morrissey, Patrick A. Ball

**Affiliations:** School of Pharmacy, University of Wolverhampton, Wulfruna Street, Wolverhampton WV1 1LY, UK

**Keywords:** antibiotic prophylaxis in dentistry, dental implant placement, pre-operative antibiotics and post-operative complications, dental implant failure

## Abstract

*Background and Objectives*: This project was developed from anecdotal evidence of varied practices around antibiotic prescribing in dental procedures. The aim of the study was to ascertain if there is evidence to support whether antibiotic (AB) use can effectively reduce postoperative infections after dental implant placements (DIPs). *Materials and Methods*: Following PRISMA-P© methodology, a systematic review of randomised controlled clinical trials was designed and registered on the PROSPERO© database. Searches were performed using PubMed^®^, Science Direct^®^ and the Cochrane© Database, plus the bibliographies of studies identified. The efficacy of prophylactic antibiotics, independent of the regimen used, versus a placebo, control or no therapy based on implant failure due to infection was the primary measured outcome. Secondary outcomes were other post-surgical complications due to infection and AB adverse events. *Results*: Twelve RCTs were identified and analysed. Antibiotic use was reported to be statistically significant in preventing infection (*p* < 001). The prevention of complications was not statistically significant (*p* = 0.96), and the NNT was >5 (14 and 2523 respectively), which indicates that the intervention was not sufficiently effective to justify its use. The occurrence of side effects was not statistically significant (*p* = 0.63). NNH was 528 indicating that possible harm caused by the use of ABs is very small and does not negate the AB use when indicated. *Conclusion*: The routine use of prophylactic antibiotics to prevent infection in dental implant placement was found to be not sufficiently effective to justify routine use. Clear clinical assessment pathways, such as those used for medical conditions, based on the patients’ age, dental risk factors, such as oral health and bone health, physical risk factors, such as chronic or long-term conditions and modifiable health determinants, such as smoking, are required to prevent the unnecessary use of antibiotics.

## 1. Introduction

A dental implant (DI) placement (DIP) procedure, with artificial root placement, is the substitution of one or more missing or defective natural teeth. It has become the definitive therapy for the restoration of entire dental arches or for partially and fully edentulous patients [1]. DI surgery can be performed in an ambulatory (outpatient) setting using local or general anaesthesia depending upon the individual preferences, health statutes and case complexity [2]. The most common fabrication materials for DIs are metals and metal alloys, polymers and ceramics [3]. Metals and metal alloys have been widely used due to advantages over other materials, including good mechanical properties, biological compatibility and excellent corrosion resistance [3,4,5]. 

To function successfully, the implant must be surrounded by healthy and stable tissues [4,5,6]. DIP has become a routine treatment option due to its high survival and success rates, offering a predictable solution for tooth replacement [7,8]. Large-scale studies reported survival rates between 97% and 75% over 10 years and 20 years, respectively [9,10]. Although its success rate is very high, failures requiring removal occasionally occur [11,12]. 

Several factors influence the DI failure rate, including poor quantity and quality of bone, inappropriate prosthesis design, inadequate number of DIs, the absence or loss of implant integration with hard and soft tissues and the practitioner’s experience. Patient-related predictors of DIP failure include an advanced age, systemic diseases, smoking, unresolved caries or infection, poor oral hygiene and parafunctional habits [13]. Lifestyle-related factors may lead to the disrupted activation of tissue healing (e.g., smoking status negatively affects the oral microbial profile and alters the microvascular environment) [14].

The DI can be made in a single step, which is called immediate loading [15]. DIP can be impacted by the presence of a bacterial load at the time of the surgery [16]. The most common organisms in endodontical procedures are *Propionibacterium acnes*, *Staphylococcus epidermidis*, *Streptococcus intermedius*, *Wolinella recta* and *Prophyromonas* and *Prevotella* species. Non-integrated or failed implants exhibit a mixed flora type of infection, such as the Gram-negative anaerobic rods *Bacteroides* spp. and *Fusobacterium* spp. [17]. Other species identified in failed implants are *Actinobacillus actinomycetemcomitans*, *Prophyromonas gingivalis* and *Prevotella intermedia* [18]. The primary criteria for assessing the successful osseointegration and survival of DIs are the implant mobility, preimplant bone loss (>1.5 mm) and the condition of the surrounding soft tissue (e.g., suppression, bleeding, pain) and infection. Patient satisfaction with the appearance, presence of discomfort and ability to function normally are used to determine the DI success [19]. 

Existing guidance is limited; prophylactic antibiotic use is not recommended in otherwise healthy individuals but may be required in those with risk factors and/or co-morbidities, but in current studies, the variations in the design and methodology used and how the outcomes were measured has made it difficult to combine or to produce clear evidence to inform the development of guidance for the appropriate use of antibiotics in DIP or the timing of their use. This review highlights these variations and limitations and provides recommendations for greater standardisation of methods and outcome measures in future studies to build a clearer base of evidence for practice in at-risk patient groups.

## 2. Materials and Methods

This systematic review aims to assess the efficacy and safety of ABs among patients undergoing DI placement procedures. The review objectives included the identification of the type of ABs used, understanding how effective ABs are in reducing or preventing infection in DIPs and what is the incidence rate of AB adverse effects. 

A systematic review was designed and implemented to investigate the efficacy and safety of ABs (independent of the type of ABs used, dose administered and timing of course of administration (preoperatively, postoperatively or both), as part of treatment, with or without AB therapy (no treatment or placebo) on DI failure and postoperative complications during DIP procedures. 

The study protocol followed the Preferred Reporting Items for Systematic Reviews and Meta-Analyses (PRISMA-P©, Ottawa Hospital Research Institute and University of Ottawa, Ottawa, Canada) statement and the Cochrane Handbook for Systematic Reviewers’ methodological guidelines (Cochrane collaboration, Oxford, UK) [20]. This systematic review was registered in the PROSPERO© (National Institute of Health, York, UK) database CRD42021269522. All statistical analyses were performed using Review Manager (RevMan^®^) statistical software version 5.4.1 (Cochrane collaboration, Oxford, UK). Risk ratios (RRs) with a 95% confidence interval (CI) were reported for dichotomous outcomes. To calculate the pooled RRs of the different studies, the Cochran-Mantel-Haenszel (M-H2) and 95% CI for effect measurement in meta-analysis was employed. Meta-analysis was used to combine outcome data from different studies using random effects or a fixed effect model [21].

The results were aggregated using the Cochran-Mantel-Haenszel (M-H2). The efficacy of the AB was assessed using the risk ratio (RR). The pooled RR with the corresponding 95% confidence interval (95% CI) was chosen as the effect size. 

The effectiveness of the treatment with ABs to prevent infection, which can lead to implant failure or implant complications, was assessed using the number needed to treat (NNT), and safety was assessed by calculating the number needed to treat to cause harm (NNH), where an NNT of 1–5 is desirable [22,23]. All decimals in the NNT have been rounded up to the nearest whole number. 

The statistical unit for “implant failure by patients”, “post-operative infection” and AB adverse events was the number of patients. The statistical unit for implant failure by implant was the number of implants. No restriction criteria were applied for the definition of each outcome reported in this review. 

### 2.1. Search Strategy

Search strategies utilised a combination of keywords using Medical Subject Headings (MeSH). No language limits were applied (date of last search: 17 January 2022). Publications included within PubMed© were searched since January 2000 until 17 January 2022. 

The following term combination was used in the PubMed©, Science Direct^®^ and Cochrane© databases: ((Dental Implant insertion OR dental implants OR dental implant OR dental implant surgery OR Dental surgery OR dental implant placement OR oral implant surgery OR implant surgery [MeSH Terms]) AND (Implant failure OR implant loss OR Implant survival OR implant survival)) AND (Antimicrobial or antibacterial or antibacterial or antibiotics or antibiotic)) AND (randomised controlled trials OR controlled clinical trial OR randomised controlled trials OR random allocation OR single-blind method OR single-blind method OR double-blind method OR clinical trial OR clinical trials OR placebos OR placebo-controlled randomised clinical trial) AND (efficacy OR Effect OR effectiveness)) AND (“Antimicrobial” or “antibiotics”, “prophylactic” OR “prophylaxis” OR “pre-operative” OR “preoperative” OR “peri operative” OR “postoperative” OR “preventative”). Issues of interest, according to the study population, intervention, comparative group, outcome measured and study design (PICOS [24]), were as follows: 

(P): Healthy adult patients over 15 yrs. Of any gender subjected to DIP.

(I): Oral or systemic AB, independent of the type of ABs used, dose administered and timing of administration (preoperatively and/or postoperatively).

(C): AB versus no AB treatment or control or placebo.

(O): Type of outcome measures: Primary outcome of interest was the post-implant infection leading to implant failure; secondary outcomes: other post-surgical complications due to infection and AB adverse events.

(S): Primary care, community or hospital setting.

### 2.2. Selection Criteria

Studies were included in the review when they were as follows:Double-blind and placebo-controlled randomised clinical trial.Patients over 15 years of age.Patients undergoing DIP procedure/s.Only oral or systemic route of AB administration.

Studies comparing the use of any AB, independent of the type of ABs used, dose administered, timing of course of administration (preoperatively, postoperatively or both; alone or no treatment or control with no AB use) following the DIP procedure.

Studies were excluded from the review when they were as follows: Systematic and meta-analysis review.Studies conducted prior to the year 2000.Comparative AB therapy with no placebo/control.Trials comparing AB therapy versus another AB.

### 2.3. Selected Studies Summary

The initial search yielded 103 studies. An additional five RCTs were identified through a manual search of the similar articles and reference lists of included studies. After an investigation of titles and then abstracts, 74 articles were considered to be ineligible for inclusion. The full texts of 34 eligible articles were then reviewed, and 22 were excluded, leaving 12 studies that accurately fit the inclusion criteria. Selected RCTs were read in full text, by two independent reviewers. A PRISMA flowchart of the screening process for the systematic review and meta-analysis was developed (Figure 1). Following a review of these articles, 12 RCTs met the criteria for inclusion in this systematic review and meta-analysis. 

A flow diagram of the included study population, reported according to Consolidated Standards of Reporting Trials (CONSORT, The EQUATOR Network and UK EQUATOR Centre, Centre for Statistics in Medicine, Nuffield Department of Orthopaedics, Rheumatology and Musculoskeletal Sciences, University of Oxford, Botnar Research Centre, Windmill Road, Oxford, OX3 7LD, UK) is shown in Figure 2.

## 3. Results 

Five corresponding authors were contacted for clarification of the study design via e-mail inquiries [25,26,27,28,29], and two authors were contacted for clarification on the number of patients with implant failure, who provided the missing data; however, three authors did not respond [26,27,28]. Table 1 and Table 2 describe the characteristics of the included studies and the reviewer comments. Six studies identified DI failure by number of patients, eight studies investigated the number of implants failed by the DI, eight assessed comparative data on postoperative complications and five reported data on the total adverse events that occurred between groups during the study period. Due to the small number of studies included in this meta-analysis, a funnel plot was not assessed [30].

Informed consent was reported in all studies except in Abu-Ta’a et al. [32] and Laskin et al. [37]. Only Caiazzo et al. [25], Esposito et al. [31], Durand et al. [29] and Esposito et al. [36] did not describe their ethical clearance or approval. Six studies reported both participant consent and ethics approvals [26,27,28,33,34,35].

One study reported preoperative antibacterial mouthwash [27], three studies reported postoperative oral hygiene [35,36,37] and six studies reported that patients had pre- and postoperative oral hygiene [25,28,29,31,32,33]. One study reported both preoperative oral hygiene and the intravenous or intramuscular administration of 4 mg of dexamethasone for the postoperative period [26]. One RCT did not report on oral hygiene [34]. Seven implemented additional therapies with anti-inflammatory and pain relief medications postoperatively [25,26,27,28,29,33,35], whilst five studies did not use additional therapies [31,32,34,36,37]. The type of analgesic used was not described in the study by Tan et al. [27]. 

Only four studies evaluated the number of supplemental analgesics between the intervention and control groups over the post-operative time points. The number of analgesics consumed in patients who were not given AB was statistically significant in the study by Nolan et al. [33]. However, the number of supplemental analgesics taken by the participants in each group was not significant in Tan et al. [27], Payer et al. [28] and Durand et al. [29]. The remaining studies did not evaluate the pain medication use [25,26,31,32,34,35,36,37]. 

Standard surgical procedures were performed in seven studies [26,27,28,29,32,33,35], whilst five did not describe their surgical protocol [25,31,34,36,37]. There were seven studies that included smokers [26,27,28,31,32,33,36]. A low level of smoking (<20) was not associated with a risk of postoperative infection and consequent implant failure in six studies. However, the study conducted by Abu-Ta’a et al. [32] included patients who were heavy smokers (>40 cigarettes/day) in their study, and the results indicated that this level of smoking affected the outcome. Five studies did not report on participants’ smoking status [25,29,34,35,37].

Five studies reported that surgery was performed by oral and maxillofacial surgeons [25,26,27,28,34]. Implantologists performed the procedures in Durand et al. [29] and odontologists in the Abu Ta’a et al. [32] study. Dentists were reported in two other studies [31,36]. However, in the Nolan et al. [33] study, the procedure was performed by two postgraduate students supervised by an experienced periodontist. Laskin et al. [37] and Khoury et al. [35] did not report on who performed the procedure. Five studies implemented a one-stage implant installation protocol without the need for a second-stage procedure to expose the implant [27,29,31,32,35]. Six studies implemented two-stage surgery [25,26,28,33,36,37]. The study by Kashani et al. [34] employed both one- and two-stage procedures. 

The trial by Payer et al. [28] was supported by a grant from the Foundation of International Team for Implantology (ITI Foundation). Funding for the trial by Anitua et al. [26] was provided by the Biotechnology Institute, Vitoria, Spain. The study by Laskin et al. [37] was USA government-supported research, and there were no restrictions on study design. The studies of Esposito et al. [36] and Esposito et al. [31] did not declare a funding source; however, the placebo and ABs used were donated by a drug company manufacturing generic preparations. These companies were not involved in the design of the study, in the data evaluation or in commenting on the manuscript. Seven studies did not report external funding [25,27,29,32,33,34,35].

No sample size calculation was reported in four studies [32,33,35,37]. Six described their sample size calculation but stated that their planned sample was not achieved leaving their studies insufficiently powered to confirm significance [25,26,28,29,31,36]. Two studies did not indicate whether their calculated sample was achieved [27,34]. All studies reported on the AB type and regimens. No study reported on the agent used as a placebo.

Six studies assessed implant failure as a main outcome in their trial. Each study reporting different definitions of implant failure. Only Nolan et al. [33] did not define “implant failure” in their study. 

Caiazzo et al. [25]: implant failure is mechanical implant removal due to the lack of osteointegration. 

Esposito et al. [31]: implant failure is an implant mobility measured manually and/or any infection dictating implant removal. The study also reported the number of patients with implant-supported prostheses, and the number of events for prosthetic failure was added to the total number of implant failure events. Prostheses failure was defined as prostheses that could not be placed or prosthesis failure (if secondary to implant failures).

Abu Ta’a et al. [32]: defined a failed implant as the presence of signs of infection and/or radiographic peri-implant radiolucencies that could not respond to a course of ABs and/or judged a failure after performing an explorative flap surgery by an experienced periodontologist.

Kashani et al. [34]: defined implant failure as the removal of an implant for any reason, from implant placement to abutment connection or prosthetic treatment.

Esposito et al. [36]: defined implant failure as implant mobility of each implant, measured manually, and/or any infection dictating implant removal. The study also reported the number of patients with implant-supported prostheses, and the number of events for prosthetic failure was added to the total number of implant failure events. The prostheses failure was defined as prostheses that could not be placed or prosthesis failure if secondary to implant failures. 

Four studies did not define postoperative complications [25,34,35,37], whilst eight evaluated the postoperative complications as a main outcome. However, in three studies, the number of events in each group was not reported and only the *p* value was stated [27,28,33]. 

Esposito et al. [31]: postoperative complications defined as any complications, such as wound dehiscence, suppuration, fistula, abscess and osteomyelitis. 

Anitua et al. [26]: the diagnosis of the presence of postoperative infections or post-implant infection been carried out using clinical criteria, which include inflammation, pain, heat, fever and discharge.

Abu Ta’a et al. [32]: postoperative infection was defined as the presence of purulent drainage (pus) or fistula in the operated region, together with pain or tenderness, localised swelling, redness and heat or fever of more than 38 °C. 

Nolan et al. [33]: different variables used to identify post-operative infection and postoperative mortality were defined as swelling, bruising, wound dehiscence and suppuration. 

Tan et al. [27]: postsurgical complications defined as flap closure, pain, swelling, suppuration and implant instability. 

Payer et al. [28]: post-surgical complication defined as pain, swelling, implant instability, purulent discharge and flap closure. 

Durand et al. [29]: postoperative surgery-associated morbidity defined as swelling, bruising(ecchymosis) and suppression.

Esposito et al. [36]: biological complications defined as any biological complications, such as wound dehiscence, suppuration, fistula, abscess and osteomyelitis. 

Three studies reported zero adverse events [26,31,32]. Three studies did not define the postoperative AB adverse events [26,29,32]. Two studies defined medication-related adverse events as any postoperative events, such as erythema multiforme, urticaria, nausea, vomiting and diarrhoea [31,36]. Other studies neither described adverse events nor reported on their prevalence [25,27,28,33,34,35,37].

### 3.1. Quality Assessment of the Selected Studies

The quality assessment tool of the Effective Public Health Practice Project (EPHPP, McMaster University 1280 Main Street West, Hamilton, ON, Canada, L8S 4K1) [38] was used to examine each study against six components. The study quality was assessed independently by two reviewers. Table 3 shows the global quality rating. All studies were classed as moderate–strong. 

### 3.2. Statistical Analysis

#### 3.2.1. Implant Failure Not Prevented by the Use of ABs, by Number of Implants

Eight studies evaluated the efficacy of AB treatment. Eight studies analysed the number of implants failed. Out of 1930 implants placed in the AB group, 23 failures were reported against 68 failures from 1566 implants inserted in the placebo group. 

As the heterogeneity estimate was low (*I*^2^ = 0%), the fixed effect model was used. The pooled estimate RR of 0.31 (95% CI, 0.20–0.48) was significant in the fixed effect model (*p* < 0.001) (Figure 3). 

#### 3.2.2. Implant Failure Not Prevented by the Use of ABs, by Patient

Six studies assessed implant failure by the number of patients in their trial, each reporting different definitions of implant failure outcomes. Only the trial conducted by Nolan et al. [33] did not define the implant failure outcome in their study. The fixed effect model was used because *I*^2^ < 25%, and the pooled estimate RR of 0.341 (95% CI, 0.22–0.52) was significant (*p* < 0.001) (Figure 4).

#### 3.2.3. Postoperative Complications Analysis

Eight clinical trials provided comparative data on the efficacy of AB treatment compared with the placebo, control or no AB treatment in patients undergoing implant placement based on postoperative complications. However, three studies did not report on the number of events in each group, with only the *p* value being stated [27,28,33]. Postoperative complications were reported in only five studies. The overall results show that there was no statistical significance difference for postoperative complications between the groups (*p* = 0.89). The fixed effect model was used because *I*^2^ < 25% giving a pooled estimate RR of 1.01 (95% CI, 0.61–1.70), with *p =* 0.96—not significant (Figure 5).

#### 3.2.4. Postoperative Antibiotic Adverse Events Analysis

Five studies reported data on the efficacy of AB treatment compared with placebo, control or no AB treatment based on total adverse events. Three studies reported zero adverse events related to the use of ABs [26,31,32]. Three did not define the postoperative AB adverse events [26,29,32]. In three studies [26,31,32], AB adverse events were appropriately assessed, but no events occurred in both groups (Figure 6). The 95% CI (0.08–5.04) is very wide, indicating a lack of confidence that AB use would affect adverse events (*p* = 0.63).

#### 3.2.5. Narrative Analysis of Other Outcomes

There was no significant difference between AB and NOAB groups in implants (*p* = 0.993) in five studies [25,31,32,34,36] or prothesis failure (*p* > 0.999) in two studies [31,36].

The terms implant success or implant survival were used interchangeably, with no specific criteria or definition given in most studies. In the study by Laskin et al. [37], the implant survival status was assessed from the time of placement to 36 months according to previous implant experience of the surgeon, implant coating, bone density; patient age, race and gender, incision type, mobility of the implant at placement and use of chlorhexidine; and health status. When implant survival rates were compared according to the patient age, it was observed that patients in all age groups who were provided preoperative AB treatment reported higher implant survival, except in the under 30 group. When implant survival rates were compared according to surgeon’s’ previous implant surgery experience, those surgeons with greater than 50 implant placements prior to the study had a slightly higher implant survival rate of 2.9% when preoperative ABs were used. In addition, a less experienced surgeon (<50 previous implant placements) had an even greater increase in the survival rate of 7.3% when preoperative ABs were used. When implant survival rates were compared according to the use of chlorhexidine in conjunction with preoperative ABs, its use improved implant survival by only 0.9% when preoperative ABs were provided, whereas without preoperative AB coverage, implant survival improved by 5.8%. Comparing both preoperative ABs and chlorhexidine to not using either, there was a substantial increase of 7.8% in implant survival [25,26,27,31,32,33,34,37]. Five studies did not specify the success rate [28,29,35,36].

The clinical diagnosis of an infected surgical wound was specified by Caiazzo et al. [25] as internal and external oedema and internal and external erythema, pain, heat and exudate. In Anitua et al. [26], the presence of infection was recorded at 3 days, 10 days, 1 month and 3 months after the implant placement. Six patients in AB and six patients in placebo groups experienced post-operative infections (not significant). Abu-Ta’a et al. [32] reported that patients self-reported infection using a mirror. One patient from the AB group and four patients in the NOAB groups developed post-operative infections, but significance was not reported. In Nolan et al. [33], post-operative morbidity (swelling, bruising, wound dehiscence and suppuration) was recorded on days 2 and 7 by the same examiner using Boolean variables: post-operative swelling and grading its severity, recorded by two independent examiners, where 0 was no swelling, 1 means mild swelling, 2 means moderate swelling and 3 means severe swelling. Post-operative bruising was significantly higher in the placebo group after 2 days, but not by day 7 post-operatively (*p* = 0.99). Two patients presented with suppuration, both of whom received a placebo preoperatively (not significant *p* = 0.49). 

Esposito et al. [36] reported complications, such as wound dehiscence, suppuration, fistula, abscess and osteomyelitis, which were measured after one week, two weeks and four months after implant placement, in which they found 11 patients in the AB group had postoperative complications vs. 13 patients in placebo group. Two flap dehiscence events occurred in the placebo group at one week, and four events were observed in the AB group (not significant *p* = 0.684). Two weeks after implant placement, one peri-implant mucositis was reported in the placebo group and one flap dehiscence in the AB group (not significant *p* = 0.1). Four months after implant placement, two cases were reported in the placebo group and one mobile implant (provoking severe pain) occurred in the AB group (not significant *p* = 0.623). Post-surgical complications reported by Tan et al. [27] include flap closure, pain, swelling, suppuration, and implant stability assessed by trained standardised examiners. Postsurgical complications were reported at one week, two weeks and one month after implant installation. Concerning the flap closure, 5% of patients in the placebo group did not achieve complete wound closure compared to 0% for the three other groups. There was no significant difference reported in all other follow-up points by all groups. Of all patients, 17% experienced postoperative pain at one week following implant placement, indicating no statistically significant difference among all AB groups (*p* > 0.05). The presence of swelling was reported in 21.4% of all subjects at week one (27% in ABs group and 17.5% in placebo). This resulted in no statistically significant difference among all groups at any time (*p* > 0.05). Payer et al. [28] performed clinical measurements at 1, 2, 4 and 12 weeks after surgery and found no significant differences in flap closure, pain, swelling, pus and implant stability of the operation site between the two treatment groups at any time over the post-operative observation period. The clinical diagnosis of postoperative surgery-associated morbidities was evaluated by Durand et al. [29] after 1, 3 and 16 weeks and 1 year. Postoperative swelling was measured using a form graded as follows: 0, no swelling; 1, mild swelling; 2, moderate swelling; and 3, severe swelling. Bruising, suppuration and wound dehiscence were evaluated dichotomously. The authors were contacted for the missing information about swelling and bruising in intervention and placebo groups, and they provided all data requested. Regarding suppuration, two participants in the AB group had suppuration at the one-week examination and were asked to rinse with 0.12% chlorhexidine gluconate twice daily for two weeks but no ABs were used. No more suppuration was reported at the three-week follow-up. There were no significant differences in postoperative morbidities between the intervention and placebo groups (*p* = 0.230).

In the study by Anitua et al. [26], patient records of smoking habits showed that 10 patients in the AB and 8 patients in the placebo group were smokers. However, the smoking habits variable was not associated with a higher risk of postoperative infection. Fifty percent of studies did not report on complications, and the other 50% reported no significant statistical differences between groups for all types of measured complications (swelling, bruising, wound dehiscence, pain and suppuration) [25,26,27,28,29,33,34,35,36,37].

No AB-related adverse events were reported in three studies [26,31,32]. In the study by Durand et al. [29], no definition of adverse events was given. Only one participant in the placebo group reported adverse events (diarrhoea) two days after implant placement. In the study by Esposito et al. (2008), one week after implant placement, one adverse event occurred in the placebo group (itching for one day) and one in the AB group (diarrhoea and somnolence); however, no significant difference was reported (*p* = 0.1), and no major complications linked to the use of ABs occurred. Seven studies did not report on AB-related adverse events [25,27,28,33,34,35,37].

In the study by Payer et al. [28], the patient self-reported outcomes used a visual analogue scale (VAS) (0–10) every day for the first two postoperative weeks. Results from mean and median VAS scores of bleedings, swelling and pain revealed no significant difference at day 14 (*p* > 0.05), but the mean VAS scores decreased over time (*p* < 0.001). The patients of Tan et al. [27] self-reported outcomes at days 1, 7 and 14. There was no statistically significant difference among study AB groups for bleeding, swelling, pain and bruising, indicating no superiority for the prophylactic regimen (*p* > 0.05). All other studies did not report on this outcome [25,26,29,31,32,33,34,35,36,37].

Nolan’s [33] patients kept records of the number of paracetamol 500 mg tablets taken for one week, and a VAS question was used for reporting. Higher postoperative VAS scores were recorded by the placebo group after two days postoperatively, and the difference between groups was significant (*p* = 0.003 at first day and *p* < 0.001 after 7 days). Durand’s [29] patients were given daily diaries to evaluate their postoperative pain severity for 1 week postoperatively, using self-administered questionnaires (VAS score). The median VAS score was 2 in the control group and 0 in the AB group. The overall median pain severity observed in both groups during the first seven days after surgery was considered mild. However, the perceived pain intensity difference in participants taking the postoperative placebo was statistically significant on the fourth day at noon (*p* = 0.047) and at night (*p* = 0.036) and on the 5th day at night (*p* = 0.036), indicating that patients in the postoperative placebo group experienced significantly more pain. However, it is worth noting that their control group experienced a longer implant surgery duration and higher number of implants inserted compared to those in their AB group. Durand et al. [29] reported that the mean number of supplemental analgesics taken by the participants in each group was 1.5 ± 4.5 tablets, and for those in the placebo group, it was 1.0 ± 2.8 tablets at the 1-week follow-up, which was not significant. 

Tan et al. [27] reported no significant difference in analgesic consumption groups. Paracetamol three times a day for two days after surgery was recommended by Payer et al. [28]. The percentage of patients who took analgesics did not differ significantly (*p* > 0.05) between the two treatment groups. The other studies did not report on this outcome [25,26,31,32,34,35,36,37].

The postoperative use of rescue ABs was reported by Kashani et al. [34]. Seven patients with 10 implants required postoperative AB treatment. Of 10 infected implants, seven healed after 10 days of ABs, whereas three implants were lost and subsequently reoperated. Six of the seven patients were in the NOAB group. Esposito et al. [36] reported that two AB-group and one placebo-group patients required postoperative ABs. All other studies did not report on this outcome [25,26,27,28,29,31,32,33,35,37].

#### 3.2.6. Risk of Bias

This was assessed using the Cochrane Collaboration Risk of Bias (RoB) assessment tool [39]. The risk of bias summary is illustrated in Figure 7 and Figure 8. Eleven studies had a low RoB [25,26,27,28,29,31,32,33,34,35,36], while one study presented a high RoB [37]. 

#### 3.2.7. Number Needed to Treat Calculation

##### To Prevent Implant Failure Due to Infection, by Implant 

NNT: 32. This is >5, which indicates that the intervention (use of prophylactic ABs) is not effective in preventing infection [22,23].

##### To Prevent Implant Failure Due to Infection by Patient

NNT: 14. The number of patients needed to treat (NNT) with antimicrobial prophylaxis to prevent one implant failure due to infection in previous studies was indicated as from 25 to 48 [40,41]. This review found that 14 patients undergoing DIP surgery need to receive ABs in order to prevent one implant failure due to infection occurring. The NNT is more than 5, which indicates that the intervention (use of prophylactic ABs) is not effective in preventing infection [22,23].

##### To Prevent Complications Due to Infection

NNT: 2523. The NNT is more than 5, which indicates that the intervention (use of prophylactic ABs) is not effective in preventing complications caused by infection [22,23].

##### To Cause Adverse Events

NNH: 528. The NNH is not a negative figure but more than 5, which means that the possible harm caused by the use of ABs exists, but it is very small and does not negate AB use when required [22,23].

## 4. Discussion

The major limitation of this systematic review is the small number of well-designed randomised controlled trials on the efficacy of ABs for DIP surgery identified. This may in part be due to this unfunded study not having the resources for wider searching or obtaining translations of non-English language publications. However, the RCTs that were included covered a range of countries and different healthcare systems and thus reflect a wide range of practices. The trials were well documented RCTs rated with a low RoB. This review provided evidence broadly supporting the administration of prophylactic ABs in those undergoing implant surgeries. When implant failure due to infection was analysed by implants, it was noted that the implants placed in the AB group were 67% more likely to survive than those when receiving placebo, indicating that implants placed in the placebo group were three times more likely to fail. In the studies, 3496 implants were placed in both groups; 23 failed in the AB groups and 68 in the placebo group. The effectiveness analysis of implant failure by implant suggests an antibiotic NNT of 32 to prevent one implant loss from occurring. This broadly agrees with the recommendations of the fourth European Association for Osseointegration Consensus 2015 [42].

When implant failure due to infection was analysed by patients, statistically beneficial results were found for the administration of ABs compared to those with controls. However, for the postoperative complications, no evidence of AB use benefit was found. Differences in clinical efficacy of Abs based on postoperative complications did not demonstrate significance due to the variability and small sample sizes of included studies. The most suggested AB was amoxicillin 2 g, but there is insufficient evidence to recommend this specific dosage. The number of adverse events occurring in those receiving ABs was not significantly different from that in those who received no ABs, confirming that ABs are safe to use when indicated. From 12 included trials, 6 reported a significant difference between ABs and placebo [29,31,33,34,35,37]. In contrast, the remaining studies showed no significant benefit [25,26,27,28,32,36]. Nolan et al. [33] reported that the number of implants placed appeared to significantly affect the osseointegration of DIPs and that when more implants were placed, the surgery duration was longer. 

A different implant system was used in all trials, and no statistically significant differences were reported for the success or failure of osseointegration. Most studies implemented a variety of single and combined additional therapies with anti-inflammatory and pain-relief medications postoperatively to manage postoperative pain following DIP surgery. The Esposito et al. [31] study found a correlation between immediate post-extractive implants and the number of failed implants and concluded that patients who received immediate post-extractive implants were more likely to fail compared with patients receiving delayed implants. The study of Nolan et al. [33] reported that longer procedures showed a modest correlation with post-operative pain (on VAS scores), interference of daily activities and the number of analgesics used. Only one study found smoking to have affected the outcome results, and heavy smoking was associated with an increased risk of implant failure. This does not rule out moderate smoking as a risk factor for failed implants. Abu Ta’a et al. [32] reported on parafunctions, finding they would increase the risk of implant failure. In this review, only one study addressed immediate post-extractive implants as a potential risk of failed implants. In most of the studies, the implant surgery was performed by oral and maxillofacial surgeons for placement of the implant. In one study by Nolan et al. [33], the implant surgical procedure was performed by two postgraduate students supervised by an experienced periodontist; however, the results of the study appear to be unaffected. Regarding the surgical protocol of one-stage and two-stage surgeries, none of the studies found significant factors associated with one- or two-stage surgeries with the failure or success of an implant. It is worth noting that emerging technologies, such as laser, photo-disinfection and airflow, etc., will contribute to infection risk reduction [new]. All studies implemented additional therapies with chlorhexidine oral rinse pre- and postoperatively or administered analgesic before and after implant surgery, which could have affected the results of the review. The studies also reported maintaining a high standard of infection control in surgical procedures. The results from this review were compared to previous studies (Table 4).

## 5. Limitations

The limitation of this systematic review is the small number of well-designed randomised controlled trials on the efficacy of ABs for DIP surgery available for review to date. 

In addition, due to the small number of studies included, the funnel plot was not used. Thus, it was not possible to objectively evaluate the publication bias in these meta-analyses. All RCTs that were included in these meta-analyses were conducted in various countries with different healthcare systems. Therefore, drawing definitive conclusions about the effects of ABs is a problem.

The study search was completed in January 2022; it is planned that a new systematic review will be conducted to cover studies published in 2022 and 2023.

## 6. Conclusions

Antibiotic use was reported to be statistically significant in preventing infection (*p* < 001), but not significant in the prevention of complications (*p* = 0.96), and the NNT was larger than 5 (14 and 2523 respectively), which indicates that the intervention is not sufficiently effective to justify its routine use. The occurrence of side effects was not significant (*p* = 0.63), and the NNH was 528, indicating that the possible harm is very small and does not negate the AB use when required. Given that the studies only evaluated healthy men and women, the study can be applied to systemically and periodontally healthy individual undergoing straightforward oral implant surgery placement under sterile conditions. It is important that future studies are rigorously designed and adequately powered to better inform future practice around the secondary outcomes and for complex patients with co-morbidities.

## Figures and Tables

**Figure 1 medicina-59-00713-f001:**
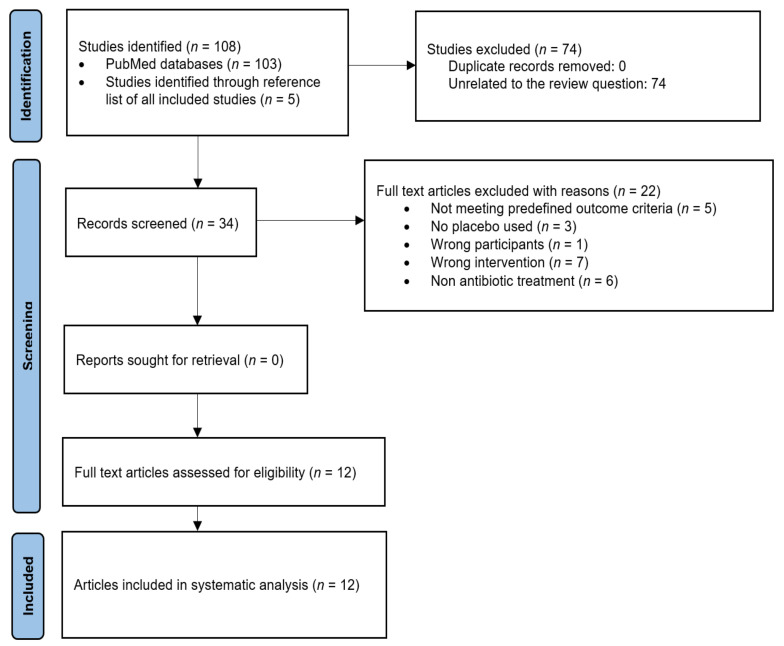
PRISMA flowchart showing the article selection process.

**Figure 2 medicina-59-00713-f002:**
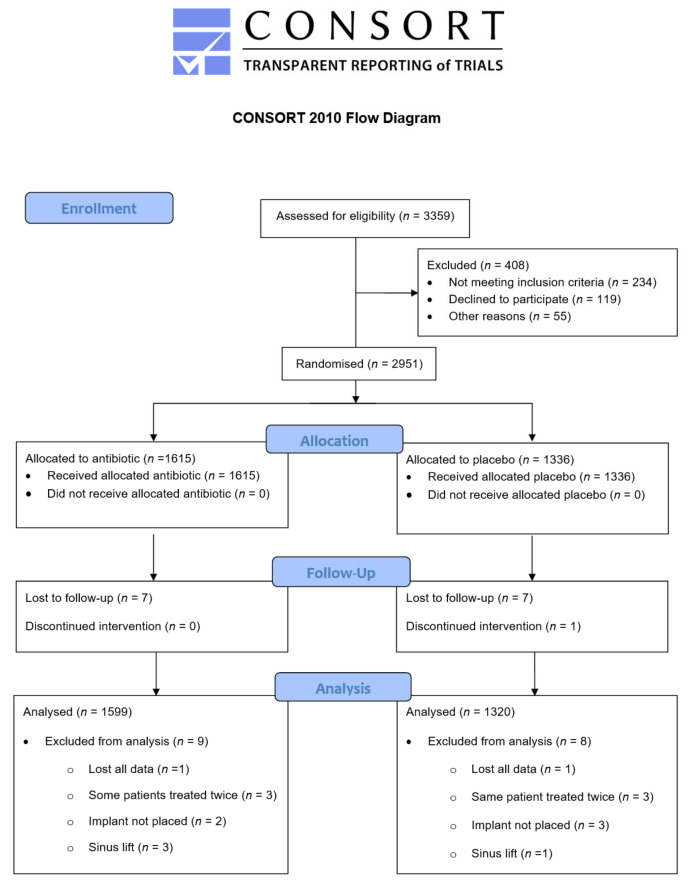
CONSORT diagram.

**Figure 3 medicina-59-00713-f003:**
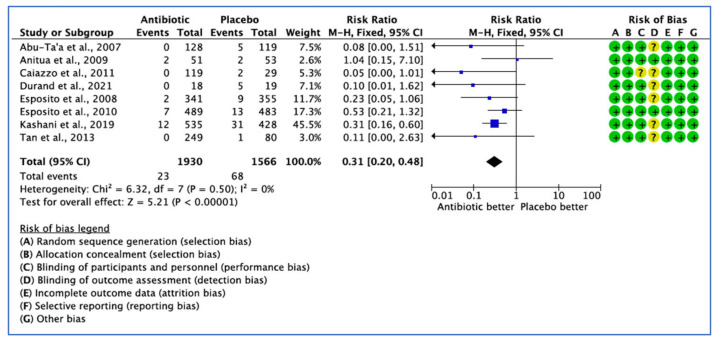
Forest plot of comparison between AB and placebo, which gives a summary estimate (centre of diamond) and its 95% confidence interval CI (width of diamond) for the number of failed implants by implant. Statistical method: Mantel-Haenszel with fixed-effect model [25,26,27,29,31,32,34,36].

**Figure 4 medicina-59-00713-f004:**
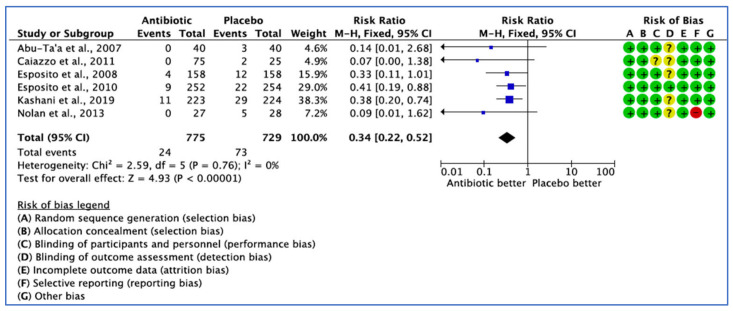
Forest plot of comparison between AB and placebo, which gives the summary estimate (centre of diamond) and its 95% confidence interval CI (width of diamond) based on implant failure by patients. Statistical method: Mantel-Haenszel with fixed-effect model [25,31,32,33,34,36].

**Figure 5 medicina-59-00713-f005:**
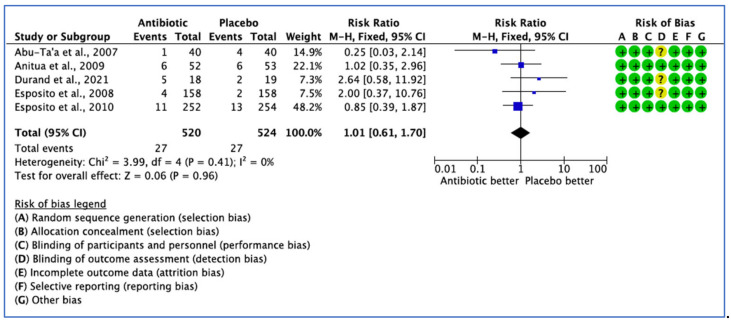
Forest plot of comparison between AB and placebo, which gives the summary estimate (centre of diamond) and its 95% confidence interval CI (width of diamond) based on postoperative complications. Statistical method: Mantel-Haenszel with fixed-effect model [26,29,31,32,36].

**Figure 6 medicina-59-00713-f006:**
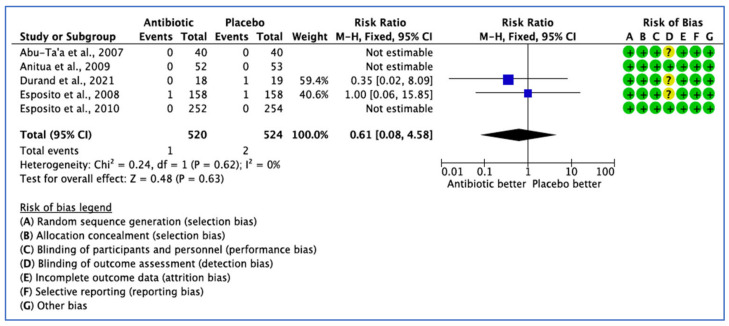
Forest plot of comparison between AB and placebo, which gives the summary estimate (centre of diamond) and its 95% confidence interval CI (width of diamond) based on postoperative adverse events. Statistical method: Mantel-Haenszel with fixed effect model [26,29,31,32,36].

**Figure 7 medicina-59-00713-f007:**
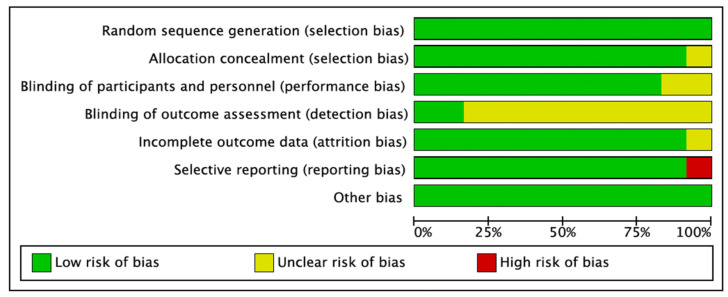
Risk of bias graph: review of authors’ judgments on each risk of bias item presented as percentages across all included studies. Green denotes features at a low risk of bias, red denotes a feature at a high risk of bias, and yellow denotes features at an unclear risk of bias.

**Figure 8 medicina-59-00713-f008:**
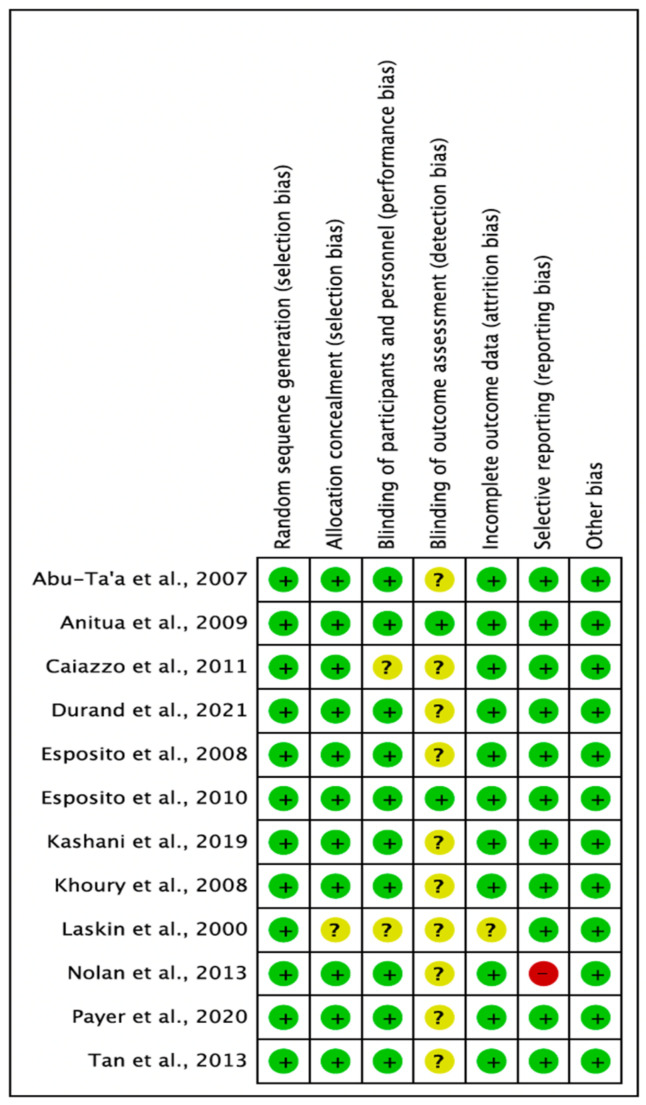
Each risk of bias item for each included study. Green denotes features at a low risk of bias, red denotes a feature at a high risk of bias, and yellow denotes features at an unclear risk of bias (Abu-Ta’a [32], Anitua [26], Caiazzo [25], Durand [29], Esposito [36], Esposito [31], Kashani [34], Khoury [35], Laskin [37], Nolan [33], Payer [28], Tan [27]).

**Table 1 medicina-59-00713-t001:** Implant studies summary.

Citation	Design	Sample Age Gender	Study Aim	Antibiotic	Control	Co-Treatment	Inflammation and Pain Relief	Antibiotic Dosing Time	Follow-Up	Study Period	Settings
Caiazzo et al., 2011 [25] Italy	Prospective, controlled, randomised clinical study Multi-centre	*n* = 100 ≥18 years F: 58 M: 42	To evaluate and compare the efficacy of different antibiotic durations in the attempt of founding the minimum effective duration of amoxicillin chemoprophylaxis in dental implant surgery.	SPAB: amoxicillin 2 g 1 h before surgery PPAB: amoxicillin 2 g 1 h before surgery and 1 g twice a day for 7 days following surgery POAB: amoxicillin 1 g twice a day started after surgery and continued for 1 week after surgery.	NOAB	Preop: chlorhexidine gluconate 0.2% solution for 1 min before each procedure Postop: chlorhexidine gluconate 0.2% twice daily for 15 days	Postoperative cold therapy (ice packs) and Nimesulide 100 mg twice daily for 3 days.	1 h before surgery	3 months	September 2006–September 2007	Dental practice
Esposito et al., 2010 [31] Italy	Double-blinded randomised controlled trial Multi-centre	*n* = 506 >18 years F: 270 M: 236	To compare the effectiveness of 2 g amoxicillin with identical placebo tablets taken orally 1 h prior to implant placement.	2 g amoxicillin	Placebo	Preop: chlorhexidine mouthwash 0.2% for 1 min prior to implant placement Postop: chlorhexidine Postoperative: chlorhexidine mouthwash 0.2% for 1 min twice a day for at least 1 week	N/R	1 h before implant placement	4 months after implant placement	April 2008– November 2009	Dental clinics
Anitua et al., 2009 [26] Spain	Double-blinded placebo-controlled randomised clinical trial Multi-centre	*n* = 105 ≥18 years F: 70 M: 35	Efficacy and safety of 2 g amoxicillin orally has been compared with identical placebo tablets 1 h before implant placement of single implants in patients with bone types II and III	2 g amoxicillin	Placebo	Preop: chlorhexidine mouthwash For 1 min Postoperative: an intravenous or intramuscular administration of 4 mg of dexamethasone, followed by decreasing doses of oral dexamethasone	Paracetamol (maximum 1 g, 8 hourly) both before and after the intervention + metamizol (575 mg, 1 or 2 tablets, 8 hourly)	1 h before implant surgery	3 months after implant placement	January 2006–September 2007	Dental clinic
Abu-Ta’a et al., 2007 [32] Belgium	Prospective, double-blinded randomised, controlled clinical trial Single centre	*n* = 80 ≥26 years F: 37 M: 43	To compare usefulness of pre- and postop antibiotics while strict asepsis was followed during periodontal surgery, investigate the effect of Preop antibiotics on the perioral skin microbial flora and to detect its impact on *S. aureus* from the noses of the patients involved.	1 g amoxicillin, 1 h Preop and 500 mg four times per day, 2 days post-operatively	NOAB	Preop: chlorhexidine di gluconate, PerioAid (0.12 solution without alcohol) for 1 min just before surgery Postop: chlorhexidine di gluconate, PerioAid (0.12 solution without alcohol twice a day for 1 min up to 7–10 days later	N/R	1 h before surgery	5 months	N/R	University
Nolan et al., 2014 [33] Ireland	Prospective, double-blind, placebo-controlled, randomised clinical trial Single centre	*n* = 55 >40 years F: 36 M: 19	To investigate the influence of pre-operative prophylactic antibiotics on post-operative morbidity: swelling, bruising, wound dehiscence and suppuration/ Postop pain and interference with daily activities and successful osseointegration of dental implants	3 g amoxicillin	Placebo	Preop: chlorhexidine 0.2% mouth rinse for at least 60 s before surgery Postop: chlorhexidine 0.2% mouthwash 4–5 times daily for the first post-operative week	Paracetamol 500 mg as required (maximum of 4 g/day)	1 h before surgery	3–4 months post-operatively	N/R	University
Tan et al., 2014 [27] Singapore Hong Kong China Australia Spain Taiwan Iceland	Single blinded RCT Multi-centre	*n* = 329 >19 years F: 147 M: 182	To determine the effects of various systemic antibiotic prophylaxis regimes on PROMs and prevalence of postsurgical complications in patients undergoing conventional implant installation	G1 (positive control, PC): 2 g amoxycillin Preop G2 (test 1, T1): 2 g Amoxycillin immediately postoperatively; G3 (test 2, T2): 2 g amoxycillin Preop, and 500 mg three times a day (8 hourly) on days 2 and 3	Placebo	Preop: 0.2 % chlorhexidine for 1 min	Description of analgesics was not provided	1 h before conventional implant placement	8 weeks	August 2009–October 2011	Dental Centre and University
Kashani et al., 2019 [34] Sweden	Prospective Double blinded RCT Multi-centre	*n* = 447 ≥15 years F: 143 M: 175	Compare the outcome of oral implant treatment due to early failures	AB group: 2 g amoxicillin or if allergic, 600 mg clindamycin 1 h pre-operatively	NOAB	N/R	N/R	1 h before surgery	7–14 days	N/R	Dental Centre and University
Payer et al., 2020 [28] Austria Singapore China Australia Hong Kong Iceland Shanghai	Double blinded RCT Multi-centre	*n* = 236 ≥21 years F: 111 M: 125	To investigate the effect of systemic antibiotics primarily on PROMs and secondarily on post-surgical complications in medically and periodontally healthy patients undergoing oral implant therapy and simultaneous GBR	2 g amoxicillin 1 h prior to implant placement and GBR + single dose of 500 mg amoxicillin 8 h after surgery and 500 mg thrice daily (8 hourly) on days 1–3 following implant placement and GBR	Placebo	Preop: 0.2% chlorhexidine for 1 min prior to surgery Postoperative: 0.2% chlorhexidine mouth rinse twice daily for 2 weeks	Paracetamol thrice daily for two days after surgery	1 h before surgery	12 weeks	August 2014–September 2017	Dental Centre and University
Durand et al., 2021 [29] Canada	Double blinded RCT Single centre	*n* = 37 Mean age 57.4/±11.3 years F: 21 M: 16	To evaluate the influence of postoperative antibiotics on peri-implant crestal bone remodelling after 1 year, postoperative pain and morbidity and one-year implant survival rate	2 g amoxicillin 1 h before implant placement followed by 7 days postoperative regimen (500 mg three times a day)	Placebo	Preop: 0.12% chlorhexidine gluconate for one minute, 1 h prior surgery Postop: 0.12% chlorhexidine twice daily for 1 week	Ibuprofen 600 mg with a maximum of four tablets per day to be taken every four hours for the first 48 h + supplemental dose of analgesic (500 mg paracetamol) taken only if needed for the first 48 h after surgery	1 h before implant placement	1 year	N/R	University
Khoury et al., 2008 [35] United states of America	Prospective observational trial—case control Single centre	*n* = 20 >18 years F: 9 M: 11	To investigate the clinical and biologic markers of early soft tissue healing around dental implants placed in the presence and absence of antibiotic prophylaxis	2 g amoxicillin, 1 h prior to surgery + 500 mg, three times a day	NOAB	Postop: 0.12% chlorhexidine rinse twice daily for 1 week	Ibuprofen 600 mg + 2 g of amoxicillin	1 h before surgery	1 week	September 2005–March 2007	University
Esposito et al., 2008 [36] Italy	Double-blinded randomised controlled trial Multi-centre	*n* = 316 ≥18 years F: 174 M: 142	To compare the efficacy of 2 g amoxicillin with identical placebo tablets taken 1 h prior to implant placement	2 g amoxicillin	Placebo	Postop: chlorhexidine mouthwash 0.2% for 1 min twice a day for at least 1 week	N/R	1 h before implant placement	4 months	September 2006–March 2007	Dental clinics
Laskin et al., 2000 [37] United states of America New Zealand	Case control Multi-centre	*n* = 702 30–90 years F: 47 M: 632	To assess the influence of preop antibiotics on long-term clinical survival of endosseous dental implants of different designs and surfaces	Penicillin 2 g	Placebo	Postop: 0.12% chlorhexidine rinse for 2 weeks	N/R	1 h before surgery	36 months	1997–2000	University Hospital

Single prophylactic antibiotic dose (SPAB), Preop and postop antibiotic treatment (PPAB), Postop antibiotic coverage (POAB), No antibiotic treatment (NOAB) N/R: Not reported, Preop: preoperative, Postop: postoperative, AB: antibiotics. Patient-reported outcome measures (PROMs), guided bone regeneration (GBR). G: group, F: female, M: male.

**Table 2 medicina-59-00713-t002:** The intervention effect of selected studies.

Citation	Main Results	Appraisal Summary
Caiazzo et al., 2011 [25]	The RCTs concluded that single preoperative, combined pre- and postoperative antibiotic and postoperative antibiotic coverage versus no antibiotic treatment did not reveal any significant differences regarding implant failure in healthy patients under normal circumstances.	This study included healthy patients. The implant type places were titanium screw-type external hex, but the implant system was not reported. The overall reported success rate was 98.65 with two failures in the NOAB group. Study limitations: The mean age included in the metanalysis was between 42- and 52-years-old, which makes the results not applicable to the global population. There was no clear explanation of the ethics approval, surgical protocol, inclusion criteria or exclusion criteria. The authors were contacted by the reviewer who requested the missing data, and they responded that ‘’the data for clinical diagnosis for the infected surgical wound was not collected’’, which can be considered as reporting bias, especially, insufficient details of the assessors blinding were also prevalent. The planned sample size could not be obtained, and reasons were not explained.
Esposito et al., 2010 [31]	Single administration of 2 g amoxicillin initiated one hour prior to implant placement could decrease early implant failure in patients not requiring a bone augmentation procedure. However, there were no significant differences for post-operative complications observed.	Implant procedures were conducted in 10 Italian private dental practices. The logistic regression model was fitted to determine confounding factors and the influence of the site effect on implant failure. No effect was observed between centres, and all had similar results (*p* = 0.97). The mean age of 49 and age range of 18–85 make the results applicable to the global population. Study limitations: No explanation provided on ethics approval or the preoperative surgical protocol. The planned sample size was 9 and could not be achieved and was insufficient to implicate a statistically significant difference.
Anitua et al., 2009 [26]	The finding does not support the use of 2 g of preoperative amoxicillin versus 2 g of placebo one hour before the placement of single dental implants in bone type II and III, and the results failed to show any advantages of preoperative antibiotics in the probability of having postoperative infection, characteristics of the saprophytic flora and incidence of antibiotic adverse events.	Single implants were placed in the maxilla or mandible. All patients received Biotechnology Institute (BTI) dental implants. Implant preparation rich in growth factor (PRGF) was applied in the operative protocol. Both researchers and patients remained blinded to the group receiving treatment. A logistic regression method was applied to determine confounding factors and the influence of different variables, including the centre, antibiotic or placebo, duration of the intervention, age and smoking habits. Study limitations: Participants had bone quality II or III; thus, the inclusion of patients with bone type I and IV could have altered the results. The planned sample size could not be achieved for a statistically significant difference. A larger sample size would be required to rule out the possibility of a difference between groups.
Abu Ta’a et al., 2007 [32]	Single dose of preoperative 1 g amoxicillin and postoperative regimen of 500 mg four times per day for 2 days does not provide any advantages concerning peri-oral aerobic and anaerobic flora on nasal aerobic and anaerobic bacteria and also failed to have a noticeable impact on post-operative infection and implant failure, while strict asepsis applies during periodontal surgery.	High-end policy of asepsis was followed for asepsis prevention from the oral and nasal cavity. Both the surgical team and the patients were blinded to the groups. A team of several peri odontologist surgeons, with different levels of experience, and nurses performed the implant surgery. Postoperative infection and implant failure were assessed at the follow-up visits for up to 5 months after the implant procedures. The implant system was not described in this study. Study limitations: Details of the enrolment period, explanation of how consent was obtained and details of the assessors blinding were missing. There were reports of patient-related variables with different conditions, such as blood-clotting problems, heavy smokers who consumed >40 cigarettes a day and patients with parafunctions included in the study, which may have affected the results.
Nolan et al., 2013 [33]	The study favoured the preoperative use of 3 g amoxicillin to improve the survival of dental implants, to effectively reduce postoperative pain, and improve interference with daily activity, considering that the duration of surgery and the number of implants had an effect on implant survival.	Only patients with the presence of a partial edentulous or edentulous alveolar ridge and presence of a tooth or several teeth regarded as non-restorable with the intention of immediate implant placement were included in the study. The number of implants in each of the study groups was not described. Multiple implant systems were used; however, 65% of implants placed were Biomet 3i. A higher level of postoperative pain following implant surgery was reported in patients with implant failure after 2 days and after 7 days. Study limitations: There was some proportion of smoker patients in antibiotic and placebo groups. The sample size was low, with only patients over the age of 40 years; therefore, the results may not be generalisable to the global population. In this study, factors related to osseointegration of the implants, such as the site of the implant and previous bone augmentation, were not described.
Tan et al., 2014 [27]	The study demonstrated that the administration of pre-, peri- or postsurgical prophylactic antibiotics did not provide beneficial effects on patient-reported outcome measures (PROMs), such as pain, swelling, bruising and bleeding, and also did not influence clinical parameters of postsurgical complications, such as flap closures, suppuration, swelling, implant stability and pain.	The study examined three antibiotic regimens and one control group using randomisation tables allocating the patient a number with a corresponding envelope. In addition, all patients were treated periodontally for one minute prior to the implant installation. The pre-operative dosage of amoxicillin applied was based on the antibiotic dosage recommendation by the AHA for the prevention of infective endocarditis. The patients were not blinded to the allocated treatment, with panned single-tooth edentulous space in the maxilla or mandible placement with adequate pristine bone for a standard oral implant placement included in this study. The reason for the implant was a single-tooth edentulous space in the maxilla or mandible with adequate pristine bone for a standard oral implant placement without the need of simultaneous bone augmentation. The implant surgery was performed by oral and maxillofacial surgeons. Only one oral implant system (Straumann SLA) was used. The oral implants were either Standard Plus or Bone Level implants and placed into pristine bone, without any simultaneous bone augmentation. Moreover, a one-stage implant installation protocol was employed. Study limitations: Light smokers who consumed fewer than 20 cigarette a day were included in the study. The lack of patient blinding, as well as descriptions of the results for postoperative complications and missing the number of implant failures, resulted in detecting reporting bias.
Kashani et al., 2019 [34]	The administration of single dose of prophylactic antibiotics in conjunction with implant placement surgery had a statistically significant lower early implant failure rate based on the early implant rate on both implant and patient levels in healthy individuals.	No preoperative oral hygiene protocol was applied in this study. Both patients and the surgeon remained blinded in the present trial. Patients were healthy and were divided into subgroups to identify potential confounding variables. Regarding the implant system, four implant brands were used: 611 implants from Nobel Biocare, 236 implants from Astra Tech, 112 implants from Straumann and implants from Ankylos. The jaw type, bone graft and number of implants placed per patient were sufficiently described. Study limitations: Details of the enrolment period and surgical protocol were not described.
Payer et al., 2020 [28]	Systemic antibiotics in comparison with a placebo group were seen to provide no improvement in patient-reported outcome measures (PROMs) and no significant reduction in post-surgical complications in healthy patients undergoing advanced oral implant surgery and simultaneous GBR.	Only periodontally and medically healthy patients with a score of 1 or 2 according to the physical status classification of the American Society of Anaesthesiologists (ASA) with defects allowing for simultaneous fixture implant placement and GBR were included. The examiners and patients remained blinded. One implant system of the Straumann Standard Plus or Straumann Bone level was applied. The number of implants inserted in each group was not described. Up to four surgeons were included to perform the clinical procedure in each centre. Payer et al. (2020) were contacted to supply the missing information through e-mail inquiries; however, the corresponding author did not respond. Study limitations: The planned sample size could not be achieved, and it was supported by a grant from the international team for implantology (ITI Foundation).
Durand et al., 2021 [29]	A single preoperative dose of 2 gr of amoxicillin one hour prior to implant placement was seen to be more beneficial than an additional postoperative antibiotic regimen for the prevention of implant complications after placing straightforward platform-switched implants for uncomplicated implant surgery in healthy patients, and the addition of amoxicillin 500 mg (three times daily) 7-day postoperatively was of no advantage or benefit on peri-implant crestal bone remodelling, post-operative morbidities and implant survival after placing straightforward platform-switched implants in healthy patients after a 1-year follow-up.	Only periodontally healthy patients or those presenting with gingivitis with adequate oral hygiene undergoing straightforward platform-switched implant placement from multiple ethnic backgrounds were included in the study. The groups were homogenous in age, sex, ethnicity, education and, smoking status. In addition, the groups were similar in surgical parameters (insertion torque, incision length, bone quality, implant location [maxilla vs. mandible]) and implant system, except two parameters: unequal distribution of implant insertion in each group, despite using random allocation, and the mean surgery duration was significantly longer in the control group. All implant surgeries were conducted in one treatment centre. The surgeons, participants and examiners were all unaware of subject allocation throughout the study. The measured outcomes were mesial and distal peri-implant crestal bone levels, postoperative pain severity and postoperative morbidity. The author was contacted to provide the missing information through e-mail inquiries; however, the corresponding authors did not respond. Study limitations: Since the sample size was low and study population was healthy, the results of this study prevent any generalisation for global populations, with comorbidities, smokers, more complex surgeries involving additional bone grafting procedures and when implant surgeries were executed by inexperienced surgeons. The duration of the enrolment period was not described. There was no clear explanation of ethical approval in this study. The decision whether to prescribe perioperative antibiotics was left to the surgeon, and this lack of randomisation could contribute to selection bias. In addition, adverse events were self-reported at the follow-up appointment and not collected on a daily basis, possibly introducing a recall bias. The sample size could not achieve statistical power.
Khoury et al., 2008 [35]	Systemic amoxicillin may have a modest effect on clinical parameters during the first postoperative week and may have a limited effect on biomarkers.	Only systemically and periodontally healthy individuals or those presenting with mild gingivitis for a single tooth implant were included in this study. The two implant systems from “Astra Tech” and “Zimmer Dental Implant” were used in this study. These two implant systems have different thread pattern designs and require different osteotomy procedures, and both systems require implant placement flush with the alveolar crest. All implant surgeries were conducted in one treatment centre. Screw-type, root-form, two-piece dental implants were inserted. The majority of implants (81%) were located in the posterior sextant, and the distribution between the maxilla and mandible was similar. The short-term evaluation of one week was chosen to determine early events occurring at the soft tissue level within the time limitation of antibiotic prescription. Study limitations: This study lacked relevant outcomes measured in this review. No specification given to who performed the surgery.
Esposito et al., 2008 [36]	2 g of preoperative amoxicillin versus 2 g of placebo one hour prior to the placement of dental implants resulted in no statistically significant differences in implant failures and postoperative complications in patients not requiring bone augmentation procedures.	Randomisation was performed using sequentially numbered, identical, opaque, sealed envelopes for concealment of allocation through the application of computer-generated restricted randomisation lists. Patients were grouped into three groups, non-smokers, light smokers (up to 10 cigarettes per day) and heavy smokers (more than 10 cigarettes per day). Dentists with extensive experience in implant treatment performed the implant surgery in each centre. Various implant systems were used. The majority of implants were manufactured by “Zimmer”. There was no apparent relevant baseline imbalance between the two groups in terms of patient characteristics (gender, age, non-smoker, duration of intervention in minutes, total number of inserted implants, patients who took postoperative antibiotics and incidence of intraoperative complications). Patients and investigators remained blinded for the entire duration of the trial. Study limitations: There was no clear explanation of ethical approval. The sample size was underpowered to detect a statistically significant difference. This study was not sponsored; however, the placebo and antibiotic used in this study were donated by a drug company manufacturing generic drugs.
Laskin et al., 2000 [37]	The study showed that the use of preoperative antibiotics significantly increased dental implant survival according to the incision type, mobility at implant placement and implant type.	The study did not compromise patients with mild and severe systematic disease. One implant system of the Spectra system was used in this study. Oral hygiene was prescribed for two weeks post-operatively. Patients were followed up from the time of placement to 36 months. Study limitations: The study did not define the inclusion and exclusion criteria and has been funded by USA Government-supported research. It was not clear who performed the implant surgery. The number of participants for whom implant failure was measured in each intervention group was missing and the method was unclear, with respect to the description of the randomisation procedure, method of sequence generation and method of allocation concealment, with inadequate reports of outcomes, causing performance bias, reporting bias and attrition bias.

**Table 3 medicina-59-00713-t003:** Effective Public Health Practice Project (EPHPP) quality assessment tool rating for individual studies.

Citation	Selection Bias	Study Design	Confounders	Blinding	Data Collection	Withdrawals & Dropouts	Global Rating
Caiazzo et al., 2011 [25]	1	1	0	2	2	1	1.17
Esposito et al., 2010 [31]	2	1	1	1	2	1	1.33
Anitua et al., 2009 [26]	1	1	1	1	2	1	1.17
Abu’-Ta’a et al., 2008 [32]	1	1	0	1	2	1	1.00
Nolan et al., 2014 [33]	2	1	0	1	2	1	1.17
Tan et al., 2014 [27]	1	1	2	2	2	1	1.50
Kashani et al., 2019 [34]	1	1	0	1	2	1	1.00
Payer et al., 2020 [28]	1	1	2	1	2	1	1.33
Durand et al., 2021 [29]	2	1	1	1	1	1	1.17
Khoury et al., 2008 [35]	2	2	0	2	2	1	1.50
Esposito et al., 2008 [36]	1	1	2	1	2	1	1.33
Laskin et al., 2000 [37]	1	2	1	3	3	1	1.83

Strong = 1, moderate = 2, weak = 3, confounders not controlled for or not stated = 0. Studies with a strong (1) or moderate (2) global mean score were deemed suitable for the systematic review. Selection bias: strong = participants very likely to be representative of the target population of greater 80% participation/moderate = participants very likely to be representative of the target population of 60–79% participation/weak = participants very likely to be representative of the target population of less than 60% participation. Study design: strong = randomised controlled trials (RCTs) or controlled clinical trials (CCTs)/moderate = cohort analytic study, a case control study, a cohort design or an interrupted time series/weak = any other method or did not state the method used. Confounders: strong = controlled for at least 80% of relevant confounders/moderate= controlled for 60–79% of relevant confounders/weak = controlled for less than 60% of relevant confounders. Blinding: strong = outcome assessor not aware of intervention status of participants AND participants not aware of research question/moderate = outcome assessor is not aware of the intervention status of participants OR the study participants are not aware of the research question/weak = outcome assessor AND participants both aware of the above or blinding is not described. Data collection methods: strong = data collection tools shown to be valid AND reliable/moderate= tools are valid, but reliability not described/weak = data collection tools not shown to be valid. Withdrawals and dropouts: strong = follow-up rate more than 80% or greater/moderate = follow-up rate of 60–79% of participants/weak = follow-up rate of less than 60% participants or withdrawal and dropouts were not described.

**Table 4 medicina-59-00713-t004:** Comparison of findings with previous systematic reviews.

Citation	Comparison
Canullo et al., 2020 [43]	This study found a statistically significant benefit of antibiotics in the reduction of early implant failure at both the implant level and patient level (RR = 0.32 [0.20, 0.51], *p* < 0.001 and RR = 0.31 [0.20, 0.49], *p* < 0.001, respectively). The reported finding was based on the fixed effect model. It revealed low risk of bias for only three of the studies included. The current meta-analysis found that the majority of studies included had a low risk of bias, except one (moderate). NNT was not analysed by Canullo et al. (2020). A strength of the current metanalysis is the inclusion of a recently published study and calculation of the NNT.
Ata-Ali, Ata-Ali and Ata-Ali, 2014 [40]	The study by Ata-Ali et al., (2014) included only four, large-sample size double-blinded placebo-controlled trails with a total of 2063 implants (including 1077 in the antibiotic treatment group and 986 in the control group). The results affirmed that antibiotic treatment provides a statistically significant beneficial reduction in implant failure (*p* = 0.003) with an NNT of 48, and it was concluded that antibiotic administration lowers the odds of implant failure by 66.9%. In contrast, failing to reach statistical significance was a reduction in the incidence of postoperative infection (*p* = 0.754). The current meta-analysis is consistent with previously published meta-analyses but included a larger number of studies regardless the sample size of studies (eight double blinded placebo-controlled trails, including a combined total of 2086 implants, including 1930 implants in the antibiotic versus 1566 in the placebo group) and revealed a statistically significant reduction in implant failure (67%). This meta-analysis used risk ratio as the measure of effect, while other meta-analyses in the literature have used the odds ratio (OR) as the primary measure. To add to the limitations, the study included RCTs assessing the effect of amoxicillin as the antibiotic treatment, whereas this meta-analysis included any type of antibiotic.
Jain, Rai, Singh and Taneja, 2020 [44]	The metanalysis implemented by Jain et al. (2020) included five RCTs rated as having a low risk of bias assessing the use of preoperative amoxicillin 1 h prior to surgery with a total of 1032 patients (514 in antibiotic and 518 in placebo groups) and 1919 implants (952 in the antibiotic group and 967 in the placebo group) for the prevention of both the number of implant failures and the number of patients with implant failures after a minimum of 3 months, from implant placement in healthy individuals, and concluded that the number of implants failed was statistically fewer in the antibiotic group, *p* = 0.006, with 0% heterogeneity. The number of patients that suffered from implant failure was significantly lower in the antibiotic group, *p* = 0.006 with 0% heterogeneity. This study is consistent with the current finding; however, our study included a larger number of randomised controlled trials including any antibiotic regardless of the timing of administration and revealed statistically significant results.
Braun, Chambrone and Khouly, 2019 [45]	The systematic review and metanalysis performed by Braun et al., (2019) included six studies with a combined total of 1162 participants and a total combined number of implants placed of 2250, assessing the efficacy of antibiotic use in the prevention of both the number of implant failures and the number of patients with implant failures after a minimum of 3 months after dental implant placement in otherwise healthy individuals receiving dental implants and suggested that the number of implants performed with the use of antibiotic had a significant effect on the survival rate (*p* = 0.005; RR, 0.35; 95% CI, 0.16 to 0.72), with an NNT of 43 for the number of implants, and the number of patients with implant failure was statistically lower in the antibiotic groups (*p* = 0.002; RR, 0.33; 95% CI, 0.16 to 0.67), with an NNT of 24 for the number of patients. The current meta-analysis is consistent but included a higher number of total participants and found a smaller NNT for both implant failure analysis by the number of implants and by patient numbers: NNT, 32 and NNT, 14. The research conducted by Braun et al. (2019) assessed antibiotic-associated adverse events and prosthetic failure as a secondary outcome. The results showed no statistically significant difference for any of the other outcome adverse events (RR, 1.00, 95% CI, 0.06 to 15.85) and prosthetic failure (*p* = 0.08; RR, 0.43; 95% CI, 0.17 to 1.11). The conclusion for the antibiotic-associated adverse events was based on the inclusion of three RCTs, while the current study included five RCTs with three studies reporting no adverse events in either group. Braun included two studies on prosthetic failure in the meta-analysis, while the current meta-analysis combined the total number of prostheses failed due to implant loss and total number of implant failures together.

## Data Availability

Not applicable.

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
