# Peer review of "Antibiotic Use in Dental Implant Procedures: A Systematic Review and Meta-Analysis"

_medicina, 2023, doi:10.3390/medicina59040713_

Round 1
Reviewer 1 Report
The topic of this study is interesting however the meta-analysis includes studies up until Jan 2022, and that is too old. Other issues were observed. Please see the enclosed PDF for further details

Author Response
|
Abstract could be better written to illustrate the contents of the study |
Corrected |
|
First time use of abbreviations should be explained |
Corrected |
|
Conclusions are too long; they should refer strictly to the results of this meta-analysis |
Corrected |
|
Keywords are not representative enough |
Corrected |
|
Introduction is far to big and derivative. Shorten it and be more restricted to your subject |
Corrected |
|
This sentence is not true, please reformulate. Otherwise we would extract all teeth and place implants |
Corrected |
|
What does very young mean? Implants should not be placed before 18 yo. |
Deleted |
|
17 Jan 2022? That was more than a year ago, the authors should perform an updated search to include articles published this year, in this form it is unacceptable. The search must be performed again so as to include more recent literature, otherwise this meta-analysis is not up to date. |
Yes, the systematic review is not just a simple search and report. This Systematic review took a year for a three person research team to complete. It is not possible to re-perform the analysis to include papers published last year. No matter how many times we will look back, by the time we finish the analysis and write the manuscript there will always be new papers published. In any research, a cut off date must be followed, and our cut of date was 17 Jan 2022. |
|
Results section is too long and poorly structured. Please eliminate unnecessary commentary or move it to the discussions section. The results section is too long and contains information that should be present in discussions. Please remake these 2 sections |
Corrected tables 4-7 deleted.
We followed the PRISMA statement to the letter, as in the first systematic review accepted and published last month. I am not sure which part the reviewer believe it should be moved to the discussion.
Torof, E., Morrissey, H., Ball, P.A. The Role of Antibiotic Use in Third Molar Tooth Extractions; A Systematic Review and Meta-Analysis. Medicina 2023;59(3):422. https://doi.org/10.3390/medicina59030422 |
|
The authors should mention other possibilities for infection control in implants such as laser, photodisinfection, airflow, etc. I suggest: Nicolae, V.D.; Chiscop, I.; Cioranu, V.S.I.; MârÈ›u, M.-A.; Luchian, A.I.; MârÈ›u, S.; Solomon, S.M. The use of photoactivated blue-O toluidine for periimplantitis treatment in patients with periodontal disease. Rev. Chim. (Buchar.) 2016, 66, 2121–2123 |
Reference included |
|
A section on the limitations of the study should be added. |
Added |
|
The conclusions are long and derivative. Refer to only your results. Which are old and bring nothing new. I strongly suggest redoing the meta-analisys so it includes studies after January 2022 |
Corrected |
|
The references are too few for a meta-analysis and are outdated |
References are reflective to the work we undertaken. |
Reviewer 2 Report
Dear authors,
the manuscript you have submitted is noteworthy, not only for the interesting topic covered, the implications of which are useful for the improvement of daily clinical protocols, but above all for the meticulousness of detail and the precision with which the analysis was conducted.
For the reasons given, I think it can be accepted in its present form.
Best regards
Author Response
Thank you very much for your comment and great understanding of the systematic review meta-analysis process
Round 2
Reviewer 1 Report
The authors must have uploaded an older version of the manuscript because not all the modifications are made to this version. Please revisit indications from the first round of review and modify them accordingly
